# Development of SNP Markers for Original Analysis and Germplasm Identification in *Camellia sinensis*

**DOI:** 10.3390/plants12010162

**Published:** 2022-12-29

**Authors:** Liubin Wang, Hanshuo Xun, Shirin Aktar, Rui Zhang, Liyun Wu, Dejiang Ni, Kang Wei, Liyuan Wang

**Affiliations:** 1Key Laboratory of Tea Biology and Resources Utilization, Ministry of Agriculture, Nature Center for Tea Improvement, Tea Research Institute Chinese Academy of Agricultural Sciences (TRICAAS), Hangzhou 310008, China; 2College of Horticulture and Forestry Science, Huazhong Agricultural University, Wuhan 430070, China

**Keywords:** tea, origin and domestication, single nucleotide polymorphism, DNA fingerprinting, variety identification

## Abstract

Tea plants are widely grown all over the world because they are an important economic crop. The purity and authenticity of tea varieties are frequent problems in the conservation and promotion of germplasm resources in recent years, which has brought considerable inconvenience and uncertainty to the selection of parental lines for breeding and the research and cultivation of superior varieties. However, the development of core SNP markers can quickly and accurately identify the germplasm, which plays an important role in germplasm identification and the genetic relationship analysis of tea plants. In this study, based on 179,970 SNP loci from the whole genome of the tea plant, all of 142 cultivars were clearly divided into three groups: Assam type (CSA), Chinese type (CSS), and transitional type. Most CSA cultivars are from Yunnan Province, which confirms that Yunnan Province is the primary center of CSA origin and domestication. Most CSS cultivars are distributed in east China; therefore, we deduced that east China (mainly Zhejiang and Fujian provinces) is most likely the area of origin and domestication of CSS. Moreover, 45 core markers were screened using strict criteria to 179,970 SNP loci, and we analyzed 117 well-Known tea cultivars in China with 45 core SNP markers. The results were as follows: (1) In total, 117 tea cultivars were distinguished by eight markers, which were selected to construct the DNA fingerprint, and the remaining markers were used as standby markers for germplasm identification. (2) Ten pairs of parent and offspring relationships were confirmed or identified, and among them, seven pairs were well-established pedigree relationships; the other three pairs were newly identified. In this study, the east of China (mainly Zhejiang and Fujian provinces) is most likely the area of origin and domestication of CSS. The 45 core SNP markers were developed, which provide a scientific basis at the molecular level to identify the superior tea germplasm, undertake genetic relationship analysis, and benefit subsequent breeding work.

## 1. Introduction

Tea is one of the most popular nonalcoholic beverages in the world. The popularity of tea is associated with its excellent taste and flavor as well as its many health benefits, such as reducing the risks of cancer, cardiovascular disease, and incidence of kidney stones; protecting teeth and bones; and its anti-obesity effect [1,2,3]. Currently, the tea plant is an important economic crop that is extensively cultivated in more than 60 countries [4]. The main tea-cultivating countries, for instance, China, India, Japan, and Kenya, have undertaken a major task involving the continuous genetic improvement of tea plants. More than 1200 tea cultivars with special traits have been developed and released for cultivation worldwide [5]. Based on this situation, a simple and accurate method is needed for the variety identification, kinship, and phylogenetic analysis of tea plants. This is very important to assure the fidelity and purity of varieties when cultivars spread and to protect the rights of breeders and consumers. For breeders, the germplasm source and pedigree information can be used to construct genetic linkage maps and to improve breeding efficiency in tea plants [6].

The tea plant (*Camellia sinensis* ) is diploid (2n = 30 chromosomes), with a genome size of about 3G [7,8]. It is a self-incompatible plant species, which greatly contributes to its high genetic diversity. In recent decades, hundreds of new tea cultivars have been bred. Compared to old varieties, many new cultivars show excellent agronomic traits, such as high production and improved stress resistance and quality. Hence, most new tea gardens have been recently created using improved clonal cultivars. Cultivated tea plants can mainly be divided into two varieties: *Camellia sinensis var. assamica* (CSA, Assam type) and *Camellia sinensis var. sinensis* (CSS, Chinese type). CSA is mainly cultivated in warm tropical areas and is processed into black tea or puer tea, whereas CSS is cultivated in a broader geographical location, mainly in high-latitude areas, for green tea production. Recent studies have shown that the divergence between CSS and CSA occurred about 0.38 to 1.54 million years ago [7]. However, their genetic diversity and differentiation, especially the original relationship between CSS and CSA, still remain a mystery.

Molecular marker technologies such as RAPD and SSR have been widely used for studying tea plants [9,10,11,12]. However, with advanced bioinformation technology, traditional molecular markers no longer meet the needs of researchers. Single-nucleotide polymorphism (SNP) markers as an efficient tool have become the main choice for this work. Compared with traditional RAPD and SSR molecular markers, SNP markers have the following advantages: (1) abundant polymorphism in the plant genome [13], such as in the rice genome, where the average density of SNP markers has been reported to be about one SNP per 232 bp [14]; (2) dimorphism, as SNP markers generally consist of only two alleles compared with SSR markers and distinguish polymorphisms by fragment size and can reduce the detection error rate [15,16]; and (3) high-throughput technology eliminates traditional molecular labeling and the tedious gel electrophoresis typing procedure and improves the detection rate [17,18]. In other plants, such as tomato and litchi, a small number of SNPs have been screened for genetic diversity analysis and population structure analysis [19]. Recently, high-throughput sequencing technology and SNP array have been applied to SNP genotyping to analyze the genetic relationships and population structure [16,20]. However, it is not practical to conduct high-throughput sequencing on a large number of individuals; therefore, it is particularly important to select a small number of core SNP markers for tea plant research.

In our previous study [20], we reported on the development of the first whole tea genome 200 K SNP array with 179,970 unique and informative SNPs. At the same time, the SNP information was used to analyze the population structure and phylogenetic tree of 142 tea cultivars from different provinces in China. In this study, we performed a more in-depth analysis of the population structure and phylogenetic tree of tea cultivars, and 45 core SNP loci were carefully selected from 117 leading Chinese clonal tea cultivars. The objectives of this study were (1) to reveal the original relationship between CSS and CSA cultivars and (2) to characterize a set of suitable SNP marker tools, i.e., a standard fingerprint identification map and parentage verification.

## 2. Results

### 2.1. Diversification and Geographical Distribution of Tea Cultivars

In total, 142 tea cultivars from different provinces in China were used to perform population structure and phylogenetic analyses in our previous report. The results of the two analyses were surprisingly consistent; i.e., all cultivars were clearly divided into three groups, namely CSA, CSS, and the transitional type. (The detailed information is shown in the Appendix A.) In this study, we further analyzed the geographic distribution of the CSA, CSS, and transitional type varieties (Figure 1). Most CSA cultivars are found in south and west China, mainly Yunnan Province, which confirms previous reports that Yunnan Province is the primary center of CSA origin and domestication [21,22], while most CSS cultivars are distributed in east China, with a clear bias towards Zhejiang and Fujian Provinces. Our data support the concept of the polyphyletic domestication of tea plants and indicate that east China (mainly Zhejiang and Fujian Provinces) is the center of CSS domestication.

### 2.2. Polymorphism and Location Distribution of 4115 SNP Loci

We selected 4115 high-quality SNPs of “poly high resolution”, “no minor homozygote”, “mono high resolution”, and “hemizygous” categories from 179,970 SNP loci. In order to visually display the position of 4115 SNP loci on chromosomes, the SNPs were mapped to chromosomes of “Shuchazao” [23]. The number of markers ranged from 221 to 401 on each chromosome, and the result is shown in Figure 2a. The results of the polymorphism analysis of 4115 markers are shown in Appendix A. The minor allele frequency (MAF) ranged from 0.018 to 0.50, with an average of 0.33, and more than 99.73% of SNP loci had values greater than 0.05 (Figure 2b). The polymorphism information content (PIC) was between 0.03 to 0.38, with an average of 0.33, and more than 88.58% of SNP loci showed high diversity, with PIC exceeding 0.250 (Figure 2c). These data show that the loci are evenly distributed across the tea plant genome, with very few missing loci and high diversity, providing a solid foundation for the subsequent screening of core markers.

### 2.3. Core SNP Loci Selection and Analyses of Polymorphism

To obtain a more convenient SNP genotyping method for tea plants in the future, we further screened 4415 SNP loci with strict screening criteria: no missing allelic data, MAF, and PIC values. In order to obtain the SNP loci by PCR amplification using the designed primers directly, 35 bp up- and downstream sequences of the SNP sites were strictly conserved within the genome of “Shuchazao”. (Primers can be designed by referring to the genome to expand the sequence in the up- and downstream regions of the markers.) Based on the criteria, all 45 SNP loci were finally selected from three SNP loci on each of the 15 chromosomes (Table 1), and the up- and downstream sequences of the 45 SNP markers are available in Appendix A.

Genetic diversity analyses were performed on the 117 tea cultivars (Appendix A) using 45 SNP loci (data shown in Table 2). The MAF were in the range of 0.475 to 0.500, the mean was 0.447, and the observed number of alleles for each SNP locus was 2. The effective number (Ne) of alleles ranged from 1.93 to 2, and the mean was 1.989. Shannon’s information index (I) was between 0.675 to 0.693, with an average of 0.690. The measured heterozygosity (Ho) ranged from 0.299 to 0.590, with an average of 0.464. The expected degree of heterozygosity (He) was in the range of 0.482 to 0.502, with an average of 0.499. The PIC was between 0.366 to 0.375, with an average of 0.374. These results indicated that these core markers exhibited high polymorphism.

### 2.4. Development and Validation Marker Tools for the Identification of Germplasm and Parentage Verification

#### 2.4.1. Identification of Germplasm and Construction of DNA Fingerprints

The probability of identity (PI) is often used as a parameter for identifying the ability of markers to distinguish breeds [10,11]. For the screening of SNP loci, each SNP locus was located on different chromosomes; the nearest locus was ~6.3 Mb (2_SNP_15 and 3_SNP_15), and all sites were considered to be independently isolated. Based on the genotyping data of the 117 tea cultivars at the 45 SNP loci, PI values were calculated in GenAlex 6.5, and the results are shown in Table 2. For the SNP loci, the PI ranged from 0.375 to 0.384, with an average of 0.376. Moreover, PI values for increasing the number of locus combinations of the 45 loci were determined (Appendix A). The combined PI value for the eight SNP markers was 3.93 × 10^−4^, which indicated that 117 tea cultivars could be distinguished by using eight markers. Eight low-PI-value markers (1_SNP_2, 1_SNP_1, 3_SNP_14, 3_SNP_15, 1_SNP_13, 2_SNP_14, 1_SNP_5, and 2_SNP_4) were selected to construct the fingerprint, and all of them had unique genotypes (Figure 3). Therefore, these eight markers can be used as primary markers to identify the germplasm of tea plants. However, considering that there are at least thousands of tea plant resources [24], the remaining markers can be used as standby markers for germplasm identification. The detailed genotypes of the 117 tea cultivars based on the 45 loci are shown in Appendix A, which can provide a reference for the identification and comparison of tea germplasm resources.

#### 2.4.2. Parentage Verification

A multi-locus probability of exclusion (PE) was calculated for the 45 markers based on their SNP type as a measurement of the parentage verification performance. Three PE types were created (Figure 4): multi-locus combination types with the probability of exclusion when the genotypes of both parents are known (termed Q1) or when the genotype of only one parent is known (termed Q2) and two putative parents unknown (termed Q3). With an increase in the number of marker combinations, the PE values also increased significantly. When the number of randomly combined markers reached 40, the PE values of the three types were 0.9997, 0.9949, and 1.0, respectively. The parentage of the 117 tea cultivars was completely confirmed by the core markers. The results suggest that the markers selected in our analysis are informative and suitable for parentage verification.

The parent–child pairs among the 117 tea cultivars were calculated using Cervus 3.0 [25]. We consulted the literature to understand the existing pedigree relationships in all tea cultivars and then verified their parents and offspring using the maximum likelihood method with 95% confidence level. Methods to infer both paternal and maternal half sibship as well as full sibship in a sample of offspring and to jointly infer the parentage of the offspring when candidate father and mother samples are also available are shown in [26].

The most likely 10 pairs of parents and offspring relationships are listed in Table 3. Interestingly, we found all of the pairs of related varieties that were perfectly matched at the 45 SNP loci. Huangmeigui, Chungui, and Mingke 1 were offspring of Huangdan, and Longjing 43 and Changye Baihao were the parents of Zhongcha 108 and Foxiang 3, respectively. Our data results are consistent with the breeding records in [27]. The existing pedigree information of Mingke1 (Tieguanyin♀ × Huangdan♂) [27] and Zhongming 7 (Zhongcha 108♀ × Longjing 43♂) [28] was very clear when Tieguanyin and Zhongcha 108 were set as the mothers of Mingke 1 and Zhongming 7, respectively. A search was conducted for the candidate fathers from the 117 tea cultivars. The results showed that Huangdan was identified as the paternal parent of Mingke1. However, our data did not indicate that Longjing 43 was the paternal parent of Zhongming 7. In addition, we found that Tieguanyin was the parent of Benshan and Xingrencha, Longjing 43 was the parent of Shuchazao, and Yungui was the parent of Aifeng.

#### 2.4.3. Analysis of the Phylogenetic Tree

To verify the ability of our core markers in the cluster analysis, a phylogenetic tree of the 117 tea cultivars was constructed based on the genetic distance. First, the groups (CSS, transitional type, and CSA) to which each cultivar belonged were clearly divided [20] and are shown together with the clustering results in Figure 5. Interestingly, individuals belonging to the CSA group were clearly differentiated from the other two groups, with only a few scattered between them. However, while some individuals were distinct between two groups, in general, CSS and the transitional type were generally not clear.

## 3. Discussion

### 3.1. New Insights into the Origin and Domestication of Chinese Teas

According to available documentation, China is the first country to cultivate and utilize tea plants [29]. Although it has been recognized that China is the origin of tea plants [30], the origin of domestication of tea plants in China has long been unilateral or controversial. A previous hypothesis stated that cultivated tea plants originated in southwest China (mainly Yunnan Province) and then spread to other parts of China and the world [12,21,31]. This hypothesis was supported by the evidence that (1) the genetic diversity of tea plants shows the highest value in original regions such as Yunnan, with a decreasing trend in regions farther away from the origin center [12]; and (2) the ancient tea plants from Yunnan Province are located at the base of the phylogenetic tree according to the resequencing data of 81 tea accessions, and strong gene flow occurs from the southwest provinces to other parts of China [23]. Meegahakumbura et al. (2016, 2018) proposed that the CSA and CSS cultivars in China were independently domesticated based on nuclear DNA microsatellite analysis [21,22]. However, the exact domestication center of CSS cultivars has not yet been confirmed. A key limitation is the lack of sufficient molecular markers for diversity evaluation [30]. The development of a 200 K SNP array largely overcomes this problem [20].

In this research, we provide rich data to support inferences about the evolutionary and domestication history of tea plants. Our data support the concept of polyphyletic domestication and indicate that southwest China (mainly Yunnan Province) is the domestication center of CSA cultivars (Figure 1). Moreover, our results also support the new idea that the formation of CSS cultivars has not likely been driven by human intervention and that east China (mainly Zhejiang and Fujian Provinces) is likely the domestication center of CSS cultivars. The genomic analyses revealed that CSA and CSS cultivars diverged around 0.38 to 1.54 million years ago, raising the suggestion that there may be a long predomestication period [7]. Furthermore, recent archaeological evidence has also shown that many well-preserved tea roots have been identified at the Tianluoshan site of Yuyao, Zhejiang, which suggests CSS tea plants were cultivated in Zhejiang about 6000 years ago [32]. Due to the environment and transportation mode at that time, it was unlikely that such artificial migration of tea plants from Yunnan to Zhejiang would be realized. Therefore, we deduce that east China (mainly Zhejiang and Fujian Provinces) is most likely the area of origin and domestication of CSS tea plants. Moreover, tea plants can clearly be divided into CSA and CSS cultivars as well as separate centers of domestication origin, which is also evidence that tea plants may have undergone adaptive evolution in contrast to other crops [33] (e.g., rice) that were subjected to strong domestication pressures.

### 3.2. Development of SNP Marker Tools: Identification of Germplasm, Parentage Verification, and Phylogenetic Analysis

Genetic diversity is the basis for the research of tea germplasm resources and genetic breeding. The research on the genetic diversity of tea plants has changed from morphological and biochemical markers to molecular markers [16]. Previously, the commonly used molecular markers were mainly SSR markers. However, compared with SSR markers, SNP markers have more obvious advantages as a new generation of molecular marker technology. Generally, SNPs are composed of two bases with low mutation frequency and high genetic stability; high-throughput data are more accurate, and the cost is low [18]. In other crops, such as tomato and cotton, core SNP molecular markers have been developed for genetic diversity and germplasm identification [34,35]. However, there are few reports on tea plants. Therefore, there is a need to develop high-quality core SNP markers for genetic diversity research of tea plants.

In this study, with the convenience of a practical application and a reduction in application costs, based on the 200 K gene chip technology developed, for the first time in tea plants, 45 high-quality SNP loci were screened from 177,790 high-quality SNP loci covering the whole genome of tea plants evenly with the following filter criteria: MAF, PIC, and the position on the chromosome. The polymorphism of 45 markers (A = 1.989, I = 0.690, Ho = 0.464, He = 0.499, and PIC = 0.374) is higher than the 60 SNP markers in a previous report (I = 0.512, Ho = 0.401, He = 0.341) [19] and higher than 86 SNP markers (I = 0.517, Ho = 0.370, He = 0.346) in [36]. Finally, DNA fingerprint construction, parentage verification, and the phylogenetic analysis of 117 tea varieties that are widely cultivated in China were performed. The results show that these markers are highly polymorphic and have strong fingerprinting power and kinship analysis capabilities. These results provide important reference data for genetic diversity analysis, germplasm identification, and genetic relationship analysis of tea plants

#### 3.2.1. Identification of Germplasm

Our SNP marker tools can effectively solve the problem of identifying tea germplasm resources. With the increase in the number of tea plant varieties, it is very important to accurately identify the uniqueness of each variety and to ensure the authenticity and purity of germplasm in the promotion of planting and scientific research. PI is an important indicator to evaluate the ability of molecular markers to discriminate germplasm [19]. In this study, the PI values for discriminative combinations were 8.05 × 10^−20^ for forty-five SNP markers and 3.93 × 10^−3^ for eight (Appendix A). These results suggest that these markers can accurately distinguish the 117 accessions. Since SNPs are dimorphic, and theoretically, 256 varieties can be distinguished by eight markers (2^8^ = 256), there is still room for improvement for germplasm identification. Moreover, taking into consideration that there are currently thousands of tea accessions worldwide [5], there is great potential for applications using these 45 SNP markers. These 45 SNP markers can be used to establish a standard fingerprint for each germplasm in the future. When an unknown germplasm is obtained, the SNP site information can be detected by biotechnology, which can be directly compared with the standard fingerprint, and then, the germplasm information can be accurately identified. In order to provide a reference for the construction of a standard fingerprint of a tea plant tree, we have published detailed information (Appendix A), such as the up- and downstream DNA sequences of the 45 loci, their positions on chromosomes, and the SNP genotypes of the 117 tea varieties.

#### 3.2.2. Parentage Verification

SNP marker tools can provide a basis for determining the genetic relationship of tea plants. Correct pedigree relationships are helpful for linkage map construction, core germplasm conservation, and breeding [10]. PE is a favorable parameter for evaluating kinship identification [37]. In this study, three types of PE (Q1, Q2, and Q3) for increasing combinations of the 45 markers had values of 100%, which showed that the parentage of the 117 tea cultivars could be accurately verified.

Seven cultivars that showed clear parentage to each other were also confirmed in this study (Table 3). For example, Zhongcha 108 was selected from the progeny of Longjing 43 that had suffered from radiation exposure [27]. Shuchazao was more likely to be an offspring of Longjing 43, and Shuchazao was bred from a local population in Anhui Province. Since Anhui and Zhejiang are adjacent, and breed exchanges are frequent, the relationship between the cultivars may be true. Our results are similar to those reported by Tan et al. (2015) [10]. Tieguanyin is an elite cultivar for making oolong tea; it has been cultivated for more than 200 years and is widely cultivated in Fujian Province [27], and it is frequently used as a crossing parent in hybridization breeding programs. Therefore, Benshan and Xingrencha were most likely the offspring of Tieguanyin through natural hybridization programs. Aifeng and Yungui were selected from the same local population at Puwen farm [27], and due to frequent natural interbreeding, they may be related.

#### 3.2.3. Phylogenetic Analysis

A sufficient number of molecular markers is an important basis for analyzing population diversity [30]. Previously, we used high-quality molecular markers to accurately distinguish tea plants [20]. The three varieties that originated from Zhejiang were classified into groups as the transitional type (S1). We examined the existing pedigree information [27] and found that Hanlv, Zhenong 12, and Zhenong 21 were a cross between the CSS and CSA groups, which indicates our previous cluster analysis was relatively accurate. Here, the core SNP markers we screened can provide some reference for the differentiation between CSA and other groups (CSS and transitional type), but they are not ideal in the differentiation between CSS and transitional groups. Although these markers were selected using strict criteria, a comparison with previous accurate classifications was difficult to achieve. Therefore, for the phylogenetic tree analysis, more markers should be selected as much as possible. Moreover, the results showed that many varieties were selected between CSA and CSS through the artificial hybrid, which reminds that we should pay attention to the conservation and collection of tea plant resources to improve the diversity of germplasm.

## 4. Materials and Methods

### 4.1. Plant Materials

A total of 142 cultivars (lines) that originated from the main tea growing provinces (Anhui, Fujian, Guangdong, Guangxi, Guizhou, Hubei, Hunan, Jiangxi, Sichuan, Taiwan, Yunnan, and Zhejiang) in China were examined. Young shoots of those cultivars (lines) were collected from the Songyang tea germplasm resource nursery in Zhejiang and the puer tea breeding center in Yunnan, China. Detailed information of the origin and the main characteristics of each cultivar (line) are available in the Appendix A.

### 4.2. SNP Detection and Selection from the Genome

The genome re-sequencing data of 139 tea accessions generated in previous study (Wang et al., 2020) were used for SNP discovery [38]. The methods of SNP calling and filtering were also described in previous study [38]. More than 218 million high-quality variants from the re-sequencing analysis were identified and used to design the 200 K AXIOM SNP array. The criteria used for the initial SNP quality analysis of the array included the following: one SNP marker for every 500 bp to ensure uniform distribution throughout the tea genome and the locations of SNPs on or close to genes. Finally, a set of 201,536 quality-filtered SNPs from 15 chromosomes of tea plants was selected and sent to the Affymetrix Bioinformatics Services (Santa Clara, CA, USA) for the final selection and design of the array. The quality of each SNP was further assessed using an in silico validation with proprietary software. The final 200 K AXIOM SNP array contained a total of 179,970 SNP markers. A panel of 142 diverse tea cultivars was genotyped for the initial validation of the 200 K AXIOM SNP array. The National Center for Biotechnology Information has received related information and assigned accession number GSE182082 to it [20]. Then, based on the 179,970 SNPs, information on 142 cultivars (lines) was used for the subsequent phylogenetic analysis.

In our previous report [20], we selected 4115 high-quality SNPs of “poly high resolution”, “no minor homozygote”, “mono high resolution”, and “hemizygous” categories from 179,970 SNP loci that had been used to construct the “LJ43×BHZ” linkage map. Therefore, the 4115 SNPs loci were used for the next step in developing genomic tools. The online web of bioinformatics (http://www.bioinformatics.com.cn/ accessed on 14 November 2022) was used for the visualization of SNP loci.

### 4.3. Core SNP Marker Selection

For convenience of scientific research and industrial applications in the future, from the 4115 SNP loci, 45 SNP markers were selected based on the following criteria: (1) SNP information was not missing in all cultivars; (2) minor allele frequency (MAF) >0.05; (3) the 35 bp up- and downstream areas of the SNP sites were completely conserved compared with the reference genome of tea plants (“Shuchazao”) [22], and the Blast tool in the *Tea Plant Genomic Variations Database* (TeaGVD) [39] (http://www.teaplant.top/teagvd accessed on 14 November 2022) was aligned to the reference genome “Shuchazao” [22]; (4) the polymorphism information content (PIC) was available. Finally, among the SNPs filtered based on the above criteria, three markers were selected on each chromosome based on the highest PIC index and 45 SNP markers evenly distributed on 15 chromosomes. Detailed genotype information of 117 tea cultivars on 45 SNP markers is shown in Appendix A.

### 4.4. Data Analysis

The SNP genotype information was converted to a software readable format (such as A/A = 1/1, T/T = 2/2, and A/T = 1/2). The minor allele frequency (MAF), gene diversity, heterozygosity, PIC, and Nei’s genetic distances for markers were calculated in PowerMarker v3.25 [40,41]. The neighbor-joining method was used to construct the phylogeny tree based on genetic distance, and the phylogenetic tree was constructed using ITOL (https://itol.embl.de/ accessed on 14 November 2022). GenAlex 6.5 was used to obtain the number of effective alleles (Ne), observed heterozygosity (Ho), expected heterozygosity (He), and Shannon’s information index (I) [42].

In the same way, the probability of identity (PI) for each marker and their combinations were calculated using GenAlex 6.5 [10,36]. PI represents the average probability of two random individuals having the same genotype, and the computation formula is as follows:PI = 2 (∑pi2) 2 − ∑pi4(1)
where pi is the frequency of the ith allele at a locus. All sites were considered to be independently isolated. For multiple loci combinations, the PI was calculated as the product of the individual locus PI values, assuming that all loci segregated independently. In this study, we took into account that all loci should belong to independent segregations at different chromosomal locations; therefore, PIsibs was not calculated [10].

The probability of exclusion (PE) values for each marker and the 45 marker combinations were calculated using GenAlex 6.5 [26]. A multi-locus combination is the exclusion probability when the genotypes of both parents are known (Q1) or when the genotype of only one parent is known (Q2), and when there are two putative parents unknown (Q3) [43].

Cervus 3.0 [25] was used to calculate the parentage analysis with exclusion and likelihood methods. The parameters were set as follows: the parents sampled, the mistyped, and the proportions of loci typed were set as 0.5, 0.01, and 0.99, respectively; the minimum typed loci, candidate fathers, and numbers for offspring were set as 13, 117, and 1000, respectively. Based on the simulation analysis, a critical LOD value at a 95% confidence level was calculated.

## 5. Conclusions

In this study, our data confirmed that Yunnan Province is a primary center of CSA origin and domestication, and we deduced that east China (mainly Zhejiang and Fujian Provinces) is most likely the area of origin and domestication of CSS. Moreover, 45 core markers were screened using strict criteria to obtain information from 179,970 SNPs. We analyzed 117 well-known tea cultivars in China with 45 core SNP markers. The 45 core SNP markers can be used as tools for the identification of germplasm, parentage verification, and phylogenetic analysis of tea plants.

## Figures and Tables

**Figure 1 plants-12-00162-f001:**
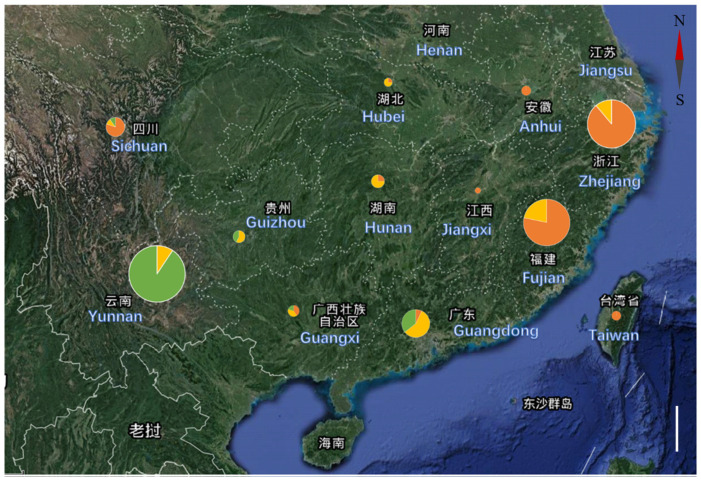
Geographical origins of tea cultivars from different provinces in China; green, Assam type (CSA); orange, Chinese type (CSS); and yellow, transitional type. The different colored areas in the pie chart represent the proportion relative to the total number of tea cultivars. Scale bar: 200 Km.

**Figure 2 plants-12-00162-f002:**
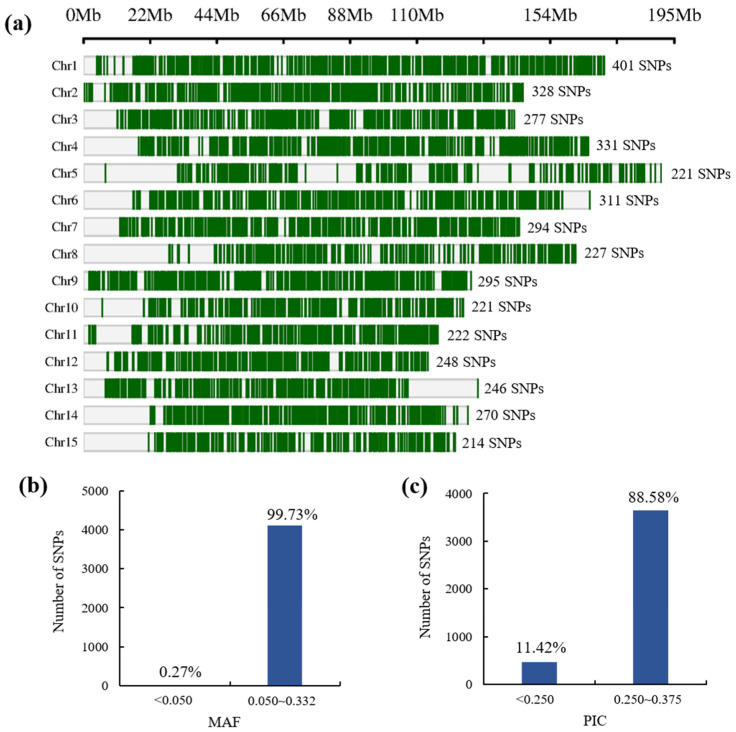
Genomic location distribution and MAF and PIC values of 4115 SNP loci: (**a**) genomic location distribution of all SNP loci; the number represents the number of markers on the chromosome; (**b**,**c**) distribution of the respective MAF and PIC values of the SNP loci.

**Figure 3 plants-12-00162-f003:**
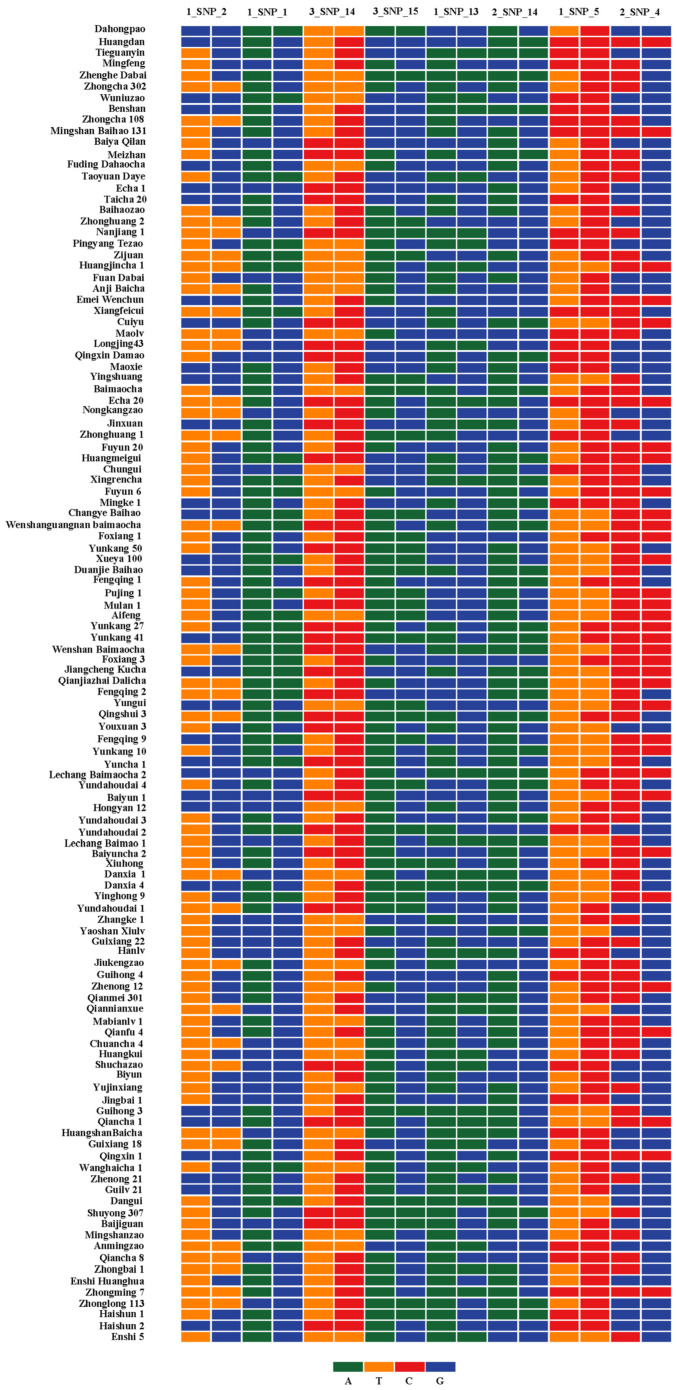
SNP fingerprinting of 117 tea cultivars based on eight SNP markers.

**Figure 4 plants-12-00162-f004:**
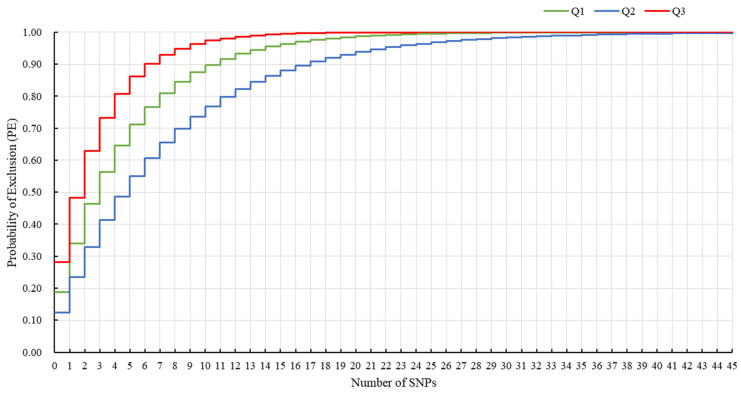
Probability of exclusion relative to the increase in the number of combinations of the 45 SNP markers. Multi-locus combination is the exclusion probability when the genotypes of both parents are known (Q1) or when the genotype of only one parent is known (Q2) and when there are two putative parents unknown (Q3).

**Figure 5 plants-12-00162-f005:**
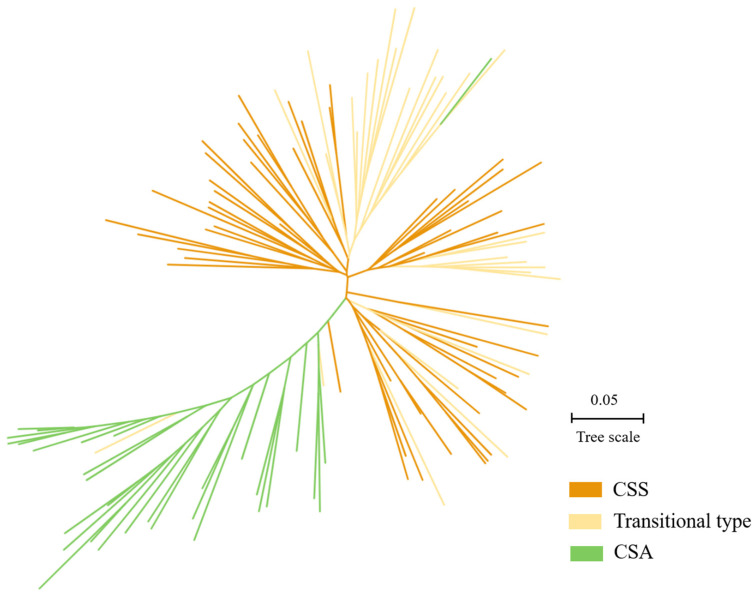
Phylogenetic analysis of the 117 tea cultivars based on the 45 SNP markers.

**Table 1 plants-12-00162-t001:** The position of the 45 SNPs’ markers on the chromosome.

Name of Marker	Chr	Position	Name of Marker	Chr	Position
1_SNP_1	1	20555681	1_SNP_9	9	97146833
2_SNP_1	1	203492932	2_SNP_9	9	14194449
3_SNP_1	1	42572356	3_SNP_9	9	33065116
1_SNP_2	2	79515402	1_SNP_10	10	120082494
2_SNP_2	2	38822259	2_SNP_10	10	162024357
3_SNP_2	2	68696654	3_SNP_10	10	14142424
1_SNP_3	3	21564028	1_SNP_11	11	103843763
2_SNP_3	3	160712162	2_SNP_11	11	120074865
3_SNP_3	3	134058046	3_SNP_11	11	115906813
1_SNP_4	4	96565277	1_SNP_12	12	157283886
2_SNP_4	4	14898710	2_SNP_12	12	84717331
3_SNP_4	4	170831355	3_SNP_12	12	119752353
1_SNP_5	5	3073649	1_SNP_13	13	110373506
2_SNP_5	5	12398569	2_SNP_13	13	68674252
3_SNP_5	5	56402023	3_SNP_13	13	103578624
1_SNP_6	6	178268444	1_SNP_14	14	33020570
2_SNP_6	6	28314239	2_SNP_14	14	52333124
3_SNP_6	6	78265333	3_SNP_14	14	102052966
1_SNP_7	7	175065396	1_SNP_15	15	85702925
2_SNP_7	7	28988576	2_SNP_15	15	108863635
3_SNP_7	7	47570898	3_SNP_15	15	102549019
1_SNP_8	8	42937619			
2_SNP_8	8	117013679			
3_SNP_8	8	43579005			

**Table 2 plants-12-00162-t002:** Genetic diversity parameters of 45 SNP loci in 117 tea cultivars.

Name of Marker	MAF	Na	Ne	I	Ho	PIC	PI
1_SNP_1	0.493	2.000	2.000	0.693	0.487	0.375	0.375
2_SNP_1	0.468	2.000	1.991	0.691	0.556	0.375	0.375
3_SNP_1	0.458	2.000	1.991	0.691	0.299	0.374	0.376
1_SNP_2	0.489	2.000	2.000	0.693	0.470	0.375	0.375
2_SNP_2	0.486	2.000	1.993	0.691	0.393	0.374	0.376
3_SNP_2	0.458	2.000	1.979	0.688	0.504	0.372	0.378
1_SNP_3	0.482	2.000	1.998	0.693	0.470	0.375	0.375
2_SNP_3	0.475	2.000	1.976	0.687	0.427	0.375	0.375
3_SNP_3	0.472	2.000	1.995	0.692	0.538	0.374	0.376
1_SNP_4	0.500	2.000	1.999	0.693	0.487	0.375	0.375
2_SNP_4	0.493	2.000	1.999	0.693	0.402	0.375	0.375
3_SNP_4	0.461	2.000	1.954	0.681	0.350	0.369	0.381
1_SNP_5	0.486	2.000	1.999	0.693	0.453	0.375	0.375
2_SNP_5	0.458	2.000	1.932	0.675	0.487	0.371	0.379
3_SNP_5	0.458	2.000	1.995	0.692	0.487	0.374	0.376
1_SNP_6	0.475	2.000	1.996	0.692	0.462	0.372	0.378
2_SNP_6	0.472	2.000	1.998	0.693	0.402	0.375	0.375
3_SNP_6	0.468	2.000	1.954	0.681	0.470	0.369	0.381
1_SNP_7	0.482	2.000	1.985	0.689	0.436	0.373	0.377
2_SNP_7	0.454	2.000	1.993	0.691	0.444	0.371	0.379
3_SNP_7	0.447	2.000	1.954	0.681	0.436	0.369	0.381
1_SNP_8	0.489	2.000	1.993	0.691	0.393	0.374	0.376
2_SNP_8	0.486	2.000	1.993	0.691	0.444	0.374	0.376
3_SNP_8	0.479	2.000	1.993	0.691	0.530	0.374	0.376
1_SNP_9	0.496	2.000	1.999	0.693	0.436	0.375	0.375
2_SNP_9	0.486	2.000	1.999	0.693	0.479	0.375	0.375
3_SNP_9	0.486	2.000	1.995	0.692	0.470	0.374	0.376
1_SNP_10	0.493	2.000	1.999	0.693	0.427	0.375	0.375
2_SNP_10	0.486	2.000	1.998	0.693	0.521	0.375	0.375
3_SNP_10	0.482	2.000	1.996	0.692	0.479	0.375	0.375
1_SNP_11	0.496	2.000	1.995	0.692	0.590	0.374	0.376
2_SNP_11	0.482	2.000	1.985	0.689	0.590	0.373	0.377
3_SNP_11	0.454	2.000	1.995	0.692	0.453	0.374	0.376
1_SNP_12	0.465	2.000	1.999	0.693	0.419	0.374	0.376
2_SNP_12	0.447	2.000	1.998	0.693	0.402	0.374	0.376
3_SNP_12	0.466	2.000	1.993	0.691	0.444	0.374	0.376
1_SNP_13	0.461	2.000	1.972	0.686	0.521	0.375	0.375
2_SNP_13	0.458	2.000	1.972	0.686	0.453	0.366	0.384
3_SNP_13	0.447	2.000	1.963	0.684	0.453	0.370	0.380
1_SNP_14	0.486	2.000	1.998	0.693	0.556	0.375	0.375
2_SNP_14	0.482	2.000	1.999	0.693	0.453	0.375	0.375
3_SNP_14	0.472	2.000	2.000	0.693	0.462	0.375	0.375
1_SNP_15	0.482	2.000	1.995	0.692	0.436	0.374	0.376
2_SNP_15	0.482	2.000	1.998	0.693	0.538	0.375	0.375
3_SNP_15	0.472	2.000	1.999	0.693	0.470	0.375	0.375
Mean	0.475	2.000	1.989	0.690	0.464	0.374	0.376

MAF, minor allele frequency; Na, number of allele; Ne, number of effective alleles; I, Shannon’s information index; Ho, observed heterozygosity; PIC, polymorphic information content; PI, probability of identity.

**Table 3 plants-12-00162-t003:** Putative parentage revealed by the fingerprinting data of the 45 SNP markers.

Offspring ^1^	Parenta ^1^	Pair Loci Mismatching	Pair LOD Scoreb	Candidate Father ID	Pair Loci Mismatching	Pair LOD Score ^2^	Known/New ^3^
Huangmeigui	Huangdan	0	9.95 *	--			Known
Chungui	Huangdan	0	8.75 *	--			Known
Mingke 1	Huangdan	0	13.5 *	Tieguanyin	0	13.53 *	Known
Benshan	Tieguanyin	0	14.3 *	--			New
Xingrencha	Tieguanyin	0	8.58 *	--			New
Zhongming 7	Zhongcha 108	0	6.2 *	Longjing 43	5	−13.32	Known
Shuchazao	Longjing 43	0	11 *	--			Known
Zhongcha 108	Longjing 43	0	10.4 *	--			Known
Foxiang 3	Changye Baihao	0	10.2 *	--			Known
Aifeng	Yungui	0	19.4 *	--			New

^1^ In each pair, which cultivar is the parent or offspring is inferred from known pedigrees or their breeding information, such as year of registration. ^2^ LOD score, the natural log of the overall likelihood ratio; a higher LOD score represents a higher chance of being the true parent according to the likelihood method. -- represents that the candidate father’s ID is unknown. * 95% level of confidence. ^3^ “Known” means that their relationship is consistent with breeding records, and “New” refers to newly discovered information based on parentage verification in this study.

## Data Availability

Not applicable.

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
