# Peer review of "Development of SNP Markers for Original Analysis and Germplasm Identification in Camellia sinensis"

_plants, 2022, doi:10.3390/plants12010162_

Round 1
Reviewer 1 Report
Plants manuscript-2118417
This a well written and tightly focused manuscript. The objectives are fully justified and are explained with detail enough to enable these SNP markers to be used widely. The subject matter will be of general interest to the readers of Plants. It also offers up avenues of understanding the breeding history and possibly the older evolution of varieties across this species large geographic range.
Specifics:
Core SNP markers implies perhaps that all the 179,970 SNP loci were evaluated and chosen as a subset for maximal explanatory (discriminatory) information. This is only partially true. There has been a hierarchical winnowing down of these loci that might itself be a useful example of developing markers in PGR if only it is explained more thoroughly. It might be best visualized as a Venn diagram that gets the reader to where the 200k goes to 179,970 goes to 4115 goes to 45.
Ln55. Emphasize for the reader that while the species is obligately outcrossing, the production fields are derived from clonally propagated ramets. It’s so obvious to the authors, but not mentioned until line 84.
Ln63. Explain why other markers are insufficient for this application. Lack of resolution? I ask because SNP markers have an inherent genomic context (LD) that is not fully exploited in this study.
Ln82. On might debate that Taiwan is a province of China…but this is not my area of expertise.
Ln89. Is there some accession identifier that can be traced to a genebank for these samples? It would seem that this would be part of any FAIR data standard.
Ln106. Give the reader a better understanding of how the SNP set was reduced to 4115 SNPs. There is no explanation, and the transition is jarring.
Ln115. Even distribution at the level of chromosomes. Could you also provide information of the distances among the three SNPs within a chromosome (the data are in the supplemental materials but would be helpful here).
Ln122-127 By “all” you are referring to the 4115 SNPs? It would seem you want ranks these variables among the 4115 SNP and pick the 45 best? This procedure has real value outside of Tea germplasm and it would be helpful to lay out the sequence of decisions. One could easily see this as a discriminant function applied iteratively to set of loci, however without any information the reader is only accepting the authors “criteria”.
Ln136. What information does expected heterozygosity really provide in a set of clonal genotypes?
Ln144. Cite a more foundational reference for Prob of Identity metrics (like Waites et al. 2001 or Paetkau et al. 1998).
Ln208 Phylogeny is not an appropriate term for describing intra-specific diversity. This is a missed opportunity for developing a more biologically meaningful estimation of population genetic of diversity in this species that allows for admixture (especially if the set of 4115 SNPS are used with LEA of something similar). Try one of two approaches (splits tree or Structure to make estimates of this differentiation. A bifurcation tree is problematic when pedigrees can (are likely) to reticulate. Even DAPC wight be a better visualization.
In general, it might be possible to tailor SNP markers around any taxonomic grouping using ordination methods like PCA for developing SNP with the most differentiation (eigen vectors). It might be helpful to describe groupings using Fst or Gst. Most importantly, treat the set of samples (142 or 117) as a population.
All figures have insufficient resolution and should be fixed. Tables and Figure should have descriptive legends that explain more details A map needs orientation, scale and a color legend (altitude?). For example, Figure 2a has an insert for SNP density…it is however not explained.
Author Response
Response to Reviewer 1 Comments
Dear Sir,
Thank you so much for your valuable comments and suggestions,we already updated all information in our new revised manuscript, the details are as follows.
Point 1: Core SNP markers implies perhaps that all the 179,970 SNP loci were evaluated and chosen as a subset for maximal explanatory (discriminatory) information. This is only partially true. There has been a hierarchical winnowing down of these loci that might itself be a useful example of developing markers in PGR if only it is explained more thoroughly. It might be best visualized as a Venn diagram that gets the reader to where the 200k goes to 179,970 goes to 4115 goes to 45.
Response 1: We used a lot of space to introduce the application value of 45 markers in scientific research and production field, so we played down the screening process of this marker, and we submitted a new version to supplement the screening details of markers.,it may make it easier for readers to comprehend the situation.
Point 2: Ln55. Emphasize for the reader that while the species is obligately outcrossing, the production fields are derived from clonally propagated ramets. It’s so obvious to the authors, but not mentioned until line 84.
Response 2: yes, it was updated. Please see the line53-line56.
Point 3: Ln63. Explain why other markers are insufficient for this application. Lack of resolution? I ask because SNP markers have an inherent genomic context (LD) that is not fully exploited in this study.
Response 3: SNP markers have many advantages campared to other markers, eg Abundant polymorphism,Dimorphism,High-throughput;the detail information showed in line 66-76. Beyond doubt, SNPs have an important application in calculating of linkage disequilibrium(LD), since we consider mainly introducing the role of 45 markers in germplasm identification, kinship verification, so, without analyzing LD.
Point 4: Ln82. On might debate that Taiwan is a province of China…but this is not my area of expertise.
Response 4: According to chinese traditions, Taiwan is a province of China.
Point 5: Ln89. Is there some accession identifier that can be traced to a genebank for these samples? It would seem that this would be part of any FAIR data standard.
Response 5: Yes, we've uploaded the sample information to NCBI, the accession number was GSE182082. We have supplemented this information in Materials and Methods, line 357: The National Center for Biotechnology Information has received related information and assigned accession number GSE182082 to it.
Point 6: Ln106. Give the reader a better understanding of how the SNP set was reduced to 4115 SNPs. There is no explanation, and the transition is jarring.
Response 6: yes, it was updated. Please see the Line112: we selected 4115 high-quality SNPs of ‘poly high resolution’, ‘no minor homozygote’, ‘mono high resolution’, and ‘hemizygous’ categories from 179,970 SNP loci.
Point 7:Ln115. Even distribution at the level of chromosomes. Could you also provide information of the distances among the three SNPs within a chromosome (the data are in the supplemental materials but would be helpful here).
Response 7: yes, it was updated. Please See the line 139
Point 8: Ln122-127 By “all” you are referring to the 4115 SNPs? It would seem you want ranks these variables among the 4115 SNP and pick the 45 best? This procedure has real value outside of Tea germplasm and it would be helpful to lay out the sequence of decisions. One could easily see this as a discriminant function applied iteratively to set of loci, however without any information the reader is only accepting the authors “criteria”.
Response 8: yes,“all” referring to the 4115 SNPs, toavoid the ambiguity,we changed “all”to 4115, see line 126.
Yes, we ranked these variables among the 4115 SNPs, then picked the 45 best. We consulted a lot of literatures then decided to employ the following method to screen markers. The detail “criteria” showed in line 370-380. (1) SNP information was not missing in all cultivars, (2) minor allele frequency (MAF) >0.05, (3) the 35 bp up- and downstream areas of the SNP sites were completely conserved compared with the reference genome of tea plants (‘Shuchazao’). After these three criteria, we got about 2000 markers from 4115 markers. Then, We computed the polymorphism of these markers. Because PIC is an essential indicator of marker polymorphism within a population, then integrated with chromosomal distribution, three markers with the highest values of PIC were chosen on each chromosome. Futhermore, our study found that 40 markers were sufficient to confirm the genetic link of 117 types and more than enough to identify 117 germplasm resources. Finally, 45 markers were obtained.
Point 9: Ln136. What information does expected heterozygosity really provide in a set of clonal genotypes?
Response 9: Expected heterozygosity(He), the computation formula is He=1-∑pi2, which is the heterozygosity calculated theoretically. Here, It only plays an auxiliary role in displaying the polymorphism information of the markers. Finally, we removed this data.
Point 10: Ln144. Cite a more foundational reference for Prob of Identity metrics (like Waites et al. 2001 or Paetkau et al. 1998)
Response 10: yes, it was updated. See line 158
Point 11:Ln208 Phylogeny is not an appropriate term for describing intra-specific diversity. This is a missed opportunity for developing a more biologically meaningful estimation of population genetic of diversity in this species that allows for admixture (especially if the set of 4115 SNPS are used with LEA of something similar). Try one of two approaches (splits tree or Structure to make estimates of this differentiation. A bifurcation tree is problematic when pedigrees can (are likely) to reticulate. Even DAPC wight be a better visualization.
Response 11 : Absolutely, I agree with you totally. Previously, we used 179,970 SNP markers to analyze all samples , we got excellent results of population structure and cluster analysis. Then, as 45 core markers were screened out, to validate whether the core markers can distinguish the population structure, we used the most commonly methods (cluster analysis and population structure, figure. 1) to analyze all samples again.. Compared with the population structure, the results of cluster analysis were more accurate. After consideration, We put the results of the cluster analysis here.
Figure. 1 The result of population structure
Point 12 :In general, it might be possible to tailor SNP markers around any taxonomic grouping using ordination methods like PCA for developing SNP with the most differentiation (eigen vectors). It might be helpful to describe groupings using Fst or Gst. Most importantly, treat the set of samples (142 or 117) as a population.
Response 12 : absolutely, I agree with you completely. We will adopt your valuable suggestion in the subsequent study.
Point 12 :All figures have insufficient resolution and should be fixed. Tables and Figure should have descriptive legends that explain more details A map needs orientation, scale and a color legend (altitude?). For example, Figure 2a has an insert for SNP density…it is however not explained.
Response 12 : yes, it was updated. Please See the line 106, line 124, line151 .

Reviewer 2 Report
This paper reports the analysis of genome-wide SNPs for their ability to assemble accurate phylogenies and genetic diversity estimates in tea. It is well-written and only needs a bit of copy editing for some formatting errors (especially different text sizes) and the English language. I am not a bioinformaticist, so I cannot comment on the specific methods that were used to identify and rank the SNPs, but I can comment on overall study design, which was excellent. The authors used an appropriate number of tea varieties from diverse geographies and whole-genome sequencing data. They filtered genome-wide SNPs to identify a small “core” set that provides the most information about genetic distance for use in rapid, routine germplasm screening. I question the utility of filtering core SNPs to focus on ones that are linked to genes, as any genes under phenotypic selection will not provide accurate estimates of genetic diversity in breeding populations. However, the authors did not require the genes near their core SNPs to be trait-linked, and they did select only those ones that provided the best genetic distance information, so it is probably okay.
This data provides a very useful toolkit for tea breeders who routinely use SNPs as molecular markers; now breeders will be able to study genetic diversity at a minimal number of loci in addition to making selections with trait-linked SNPs.
Author Response
Dear Sir,
Thank you so much for your valuable comments and suggestions,we already updated all information in our new revised manuscript.
